# BCG Provides Short-Term Protection from Experimental Cerebral Malaria in Mice

**DOI:** 10.3390/vaccines8040745

**Published:** 2020-12-09

**Authors:** Julia Witschkowski, Jochen Behrends, Roland Frank, Lars Eggers, Linda von Borstel, David Hertz, Ann-Kristin Mueller, Bianca E. Schneider

**Affiliations:** 1Junior Research Group Coinfection, Priority Area Infections, Research Center Borstel-Leibniz Lung Center, 23845 Borstel, Germany; witschkowski.julia@gmail.com (J.W.); lars.eggers@fz-borstel.de (L.E.); lvonborstel@fz-borstel.de (L.v.B.); dhertz@fz-borstel.de (D.H.); 2Core Facility Fluorescence Cytometry, Research Center Borstel-Leibniz Lung Center, 23845 Borstel, Germany; jbehrends@fz-borstel.de; 3Centre for Infectious Diseases, Parasitology Unit, Heidelberg University Hospital, 69120 Heidelberg, Germany; roland.frank81@gmx.de (R.F.); mail@annkristinmueller.de (A.-K.M.); 4German Center for Infection Research (DZIF), TTU Malaria, 69120 Heidelberg, Germany

**Keywords:** BCG, *Plasmodium berghei* ANKA, experimental cerebral malaria, heterologous immunity, non-specific protection

## Abstract

Clinical and experimental evidence suggests that the tuberculosis vaccine BCG offers protection against unrelated pathogens including the malaria parasite. Cerebral malaria (CM) is the most severe complication associated with *Plasmodium falciparum* infection in humans and is responsible for most of the fatalities attributed to malaria. We investigated whether BCG protected C57BL/6 mice from *P. berghei* ANKA (PbA)-induced experimental CM (ECM). The majority of PbA-infected mice that were immunized with BCG showed prolonged survival without developing clinical symptoms of ECM. However, this protective effect waned over time and was associated with the recovery of viable BCG from liver and spleen. Intriguingly, BCG-mediated protection from ECM was not associated with a reduction in parasite burden, indicating that BCG immunization did not improve anti-parasite effector mechanisms. Instead, we found a significant reduction in pro-inflammatory mediators and CD8^+^ T cells in brains of BCG-vaccinated mice. Together these data suggest that brain recruitment of immune cells involved in the pathogenesis of ECM decreased after BCG vaccination. Understanding the mechanisms underlying the protective effects of BCG on PbA-induced ECM can provide a rationale for developing effective adjunctive therapies to reduce the risk of death and brain damage in CM.

## 1. Introduction

*Mycobacterium (M.) bovis* bacillus Calmette–Guérin (BCG) is a live attenuated strain of *M. bovis* and the only licensed vaccine against tuberculosis (TB). It is one of the most widely used of all current vaccines, reaching >80% of neonates and infants in countries where it is part of the national childhood immunization program [1]. While BCG confers protection from the severe forms of TB in children, such as TB meningitis and miliary disease, protection against pulmonary TB in adolescents and adults is considerably variable [2]. Importantly, there is evidence that BCG may protect immunized infants from pathogens other than *M. tuberculosis*, called heterologous or non-specific protection [3,4,5,6,7,8]. As such, BCG may affect the outcome of several major infectious diseases including malaria [9,10], a hypothesis supported by animal studies, which demonstrate that BCG indeed provides non-specific protection from infection with different rodent Plasmodium strains [11,12,13,14,15,16,17]. It has been suggested that the non-specific protective effects of BCG are due to trained immunity, which increases the responsiveness of innate immune cells to other pathogens and is mediated by epigenetic changes [18,19,20,21,22]. A recent randomized, controlled phase 1 clinical trial evaluated the induction of immunity and protective efficacy of BCG against a controlled human malaria infection in 20 healthy volunteers. BCG-vaccinated volunteers had significantly earlier expression of NK cell activation markers and a trend towards earlier phenotypic monocyte activation [23]. Moreover, parasitemia was reduced in a subgroup of BCG-vaccinated volunteers and inversely correlated with NK cell and monocyte activation. The study demonstrates that BCG vaccination alters the clinical and immunological response to malaria, and emphasizes to further explore its potential in strategies for clinical malaria vaccine development.

Cerebral malaria (CM) is the most severe complication associated with *Plasmodium falciparum* infection in humans and is responsible for most of the fatalities attributed to malaria [24]. Presumably, the sequestration of parasitized erythrocytes together with overwhelming inflammatory reactions are responsible for the induction of the fatal pathology [25]. Protection against severe malaria is acquired by only one or two natural infections. Consequently, in areas where malaria transmission is intense, CM principally occurs in children under five and is rare in adults, while in low-transmission regions, CM principally occurs in adults. Despite effective antimalarial treatment, mortality is high and survivors may suffer from long-term neurological impairments [26,27]. Hence, there is an urgent need to find new strategies for the prevention of this fatal condition.

Experimental studies have shown that the modulation of parasite burden protects against severe disease [28,29,30,31,32]. Because BCG was reported to limit parasitemia in mice and man [11,12,17,23], we employed a well-characterized mouse-model of experimental CM (ECM) to find out whether BCG can mediate protection from ECM.

## 2. Materials and Methods

### 2.1. Ethics Statement and Mice

Animal experiments were approved by the Ethics Committee for Animal Experiments of the Ministry of Energy, Agriculture, the Environment, Nature and Digitalization of the State of Schleswig-Holstein (Kommission für Tierversuche/Ethik-Kommission des Landes Schleswig-Holstein) under the license 26–3/18 (“Unspezifischer Schutz einer BCG-Vakzinierung gegen eine durch PbANKA-Infektion induzierte experimentelle zerebrale Malaria (ECM)”/“Unspecific protection of a BCG vaccination against a PbANKA induced experimental cerebral malaria (ECM)”). All mice used were bred in-house under specific-pathogen-free conditions and maintained under specific barrier conditions in the BSL-2 facility at the Research Center Borstel. Female C57BL/6 mice aged between 10–16 weeks were used.

### 2.2. BCG Vaccination

*M. bovis* BCG Pasteur was grown in Middlebrook 7H9 broth (BD Biosciences, San Diego, CA, USA) supplemented with 0.05% *v*/*v* Tween 80 and 10% *v*/*v* OADC (Oleic acid, Albumin, Dextrose, Catalase). Bacterial cultures were harvested at logarithmic growth phase (OD_580_ 0.6–0.8) and aliquots were stored at −80 °C until later use. Viable cell counts in thawed aliquots were determined by plating serial dilutions of cultures onto Middlebrook 7H11 agar plates followed by incubation at 37 °C. Prior to use, bacteria were resuspended in phosphate buffered saline (PBS) and homogenized by mixing the suspension using a 27 G cannula ten times. Mice were vaccinated subcutaneously (s.c.) with 1 × 10^7^ BCG in 0.1 mL of PBS. Where indicated, heat treatment was performed at 95 °C for 30 min. Bacterial loads were evaluated by mechanical disruption of organs in 0.05% *v*/*v* Tween 20 in PBS containing a proteinase inhibitor cocktail (Roche, Basel, Switzerland) prepared according to the manufacturer’s instructions. Tenfold serial dilutions of organ homogenates in sterile water/1% *v*/*v* Tween 80/1% *w*/*v* albumin were plated onto Middlebrook 7H11 agar plates and incubated at 37 °C. Colonies were enumerated after 3–4 weeks.

### 2.3. Parasites and Sporozoite Isolation

All experiments were carried out with the rodent parasite *Plasmodium berghei* ANKA (PbA). PbA sporozoites (SPZ) were isolated by dissection of salivary glands from female Anopheles stephensi mosquitoes at day 21 after blood meal infection.

### 2.4. Parasitic Infection and Evaluation of Disease

Experimental naïve or BCG-vaccinated mice were infected intravenously (i.v.) with 1 × 10^4^ PbA SPZ or infected red blood cells (iRBC). Parasitemia was determined on Giemsa-stained blood smears from tail blood. Mice were monitored daily from day 6 until day 12 post PbA infection for clinical ECM evaluation according to the Rapid Murine Coma and Behavioral Scale (RMCBS; [33]). ECM was reported in mice that showed symptoms of severe neurological disease, defined as a RMCBS < 5. Animals were euthanized to avoid unnecessary suffering and the time point that followed was denoted the time of death.

### 2.5. Assessment of Blood Brain Barrier Integrity by Evans Blue

Mice were injected i.v. with 0.1 mL of 2% Evans blue (Sigma-Aldrich, St. Louis, MO, USA) in saline at day 8 post infection. Mice were sacrificed 1 h after injection and perfused intracardially with 20 mL PBS to remove dye from the vasculature. Brains were removed to assess the coloration.

### 2.6. Cell Isolation and Purification from Brain, Spleen and Blood

Mice were sacrificed at different time points post PbA infection. Blood was taken from vena cava and transferred into EDTA-tubes, and mice were perfused intracardially with 20 mL PBS to remove remaining circulating leukocytes from the tissue. Brains and spleens were removed and passed through a 100 µm pore size cell strainer to obtain single cell suspensions. Brains were further passed through a 70 µm pore size cell strainer. Remaining erythrocytes were lysed (155 mM NH_4_Cl, 10 mM KHCO_3_, 0.1 mM EDTA in H_2_O) and cells were resuspended in RPMI 1640 supplemented with 2 mM glutamine, 1% *v*/*v* HEPES, 50 µM β-mercaptoethanol and 10% *v*/*v* heat-inactivated fetal calf serum (complete RPMI 1640 medium). Cell numbers were determined with the Scepter^TM^ 2.0 Cell Counter (Merck, Darmstadt, Germany).

### 2.7. Flow Cytometry

To analyze surface expression of CD45, CD8, CD44, CD62L, CD160 and PD1, single-cell suspensions were incubated with fluorescently labelled antibodies (BD Biosciences, San Diego, CA, USA and Biolegend, San Diego, CA, USA) for 30 min at 4 °C in the dark and subsequently analyzed by flow cytometry on a LSR^TM^ II flow cytometer (BD Biosciences) equipped with a 405 nm, 488 nm and 633 nm laser. Data were analyzed with the FCSExpress software (DeNovo^TM^ Software, Pasadena, CA, USA).

### 2.8. Multiplex Assay

Chemokine protein levels in brain homogenates were determined by bead-based immunoassay (LEGENDplex^TM^ Mouse Proinflammatory Chemokine Panel, BioLegend) according to the manufacturer’s protocol.

### 2.9. RNA Isolation and RT-PCR

Total RNA from liver or brain tissue was extracted from peqGOLD Trifast™ using the Direct-zol™ RNA MiniPrep Kit (Zymo Research, Irvine, CA, USA) according to the manufacturer’s protocol. Isolated RNA was reverse transcribed using Maxima First Strand cDNA Synthesis Kit for RT-qPCR (Thermo Fisher Scientific, Waltham, MA, USA) according to the manufacturer’s instruction. Quantitative RT-PCR was performed using LightCycler^®^ 480 SYBR Green I Master (Roche) on the LightCycler^®^ 480 instrument. Analysis of the relative changes was performed using LightCycler480 Software 1.5.0 SP4 (Version 1.5.0.39, Roche). All quantifications were normalized to the level of GAPDH gene expression. The following primers were used:*Gapdh* forward ATTGTCAGCAATGCATCCTG; *Gapdh* reverse ATGGACTGTGGTCATGAGCC;*Pb*A *18S rRNA* forward AAGCATTAAATAAAGCGAATACATCCTTAC; *Pb*A *18S rRNA* reverse GGAGATTGGTTTTGACGTTTATGTG;*Ifng* forward TCAAGTGGCATAGATGTGGAAGAA; *Ifng* reverse TGGCTCTGCAGGATTTTCATG:*Tnf* forward CCACCACGCTCTTCTGTCTAC; *Tnf* reverse AGGGTCTGGGCCATAGAACT;*Lfa* forward CTCCAGGAGGACAACTCAGC; *Lfa* reverse CTAGTGTGGGCATGTTGTGG;*Il10* forward GAGGTGAGTGGCTGTCTGTG; *Il10* reverse CAGAGAAGCATGGCCCAGAA*Icam1* forward GGGACCACGGAGCCAATT; *Icam1* reverse CTCGGAGACATTAGAGAACAATGC;*Vcam1* forward ACAAGTCTACATCTCTCCCAGGA; *Vcam1* reverse AAGGTGAGGGTGGCATTTCC;*CD8a* forward CAAATGTCCCAGGCCGCTA; *CD8a* reverse GAACTGTCCCCATCACACCC;*Gzmb* forward GGACAAAGGCAGGGGAGATC; *Gzmb* reverse TCACAGTGAGCAGCAGTCAG.

### 2.10. Statistical Analysis

All data were analyzed using GraphPad Prism 5 (GraphPad Prism, San Diego, USA). Outliers were identified by Grubbs Outlier test with an α-value ≤ 0.05. Statistical analysis was performed by unpaired Student’s *t*-test or log rank test as described in the figure legends. Values of * *p* ≤ 0.05, ** *p* ≤ 0.01 and *** *p* ≤ 0.001 were considered significant.

## 3. Results

### 3.1. BCG Partly Protects C57BL/6 Mice from P. berghei ANKA Induced ECM

A well-characterized model of ECM uses Plasmodium berghei ANKA (PbA) infection of C57BL/6 mice, which develop severe neurological symptoms and usually succumb to infection within 6–9 days [33,34,35]. To investigate whether BCG can mediate protection against ECM, we vaccinated mice with 10^7^ BCG s.c. and infected them i.v. with 10^4^ PbA sporozoites (SPZ) at different time points as indicated in Figure 1A. All animals developed blood-stage infection. However, the majority of mice that were PbA infected early after BCG vaccination (after 10 or 30 days, respectively) showed prolonged survival without developing clinical symptoms of ECM (assessed by the Rapid Murine Coma and Behavior Scale [33]), whereas 80–100% of non-vaccinated mice succumbed to ECM 8 to 10 days after challenge (Figure 1B,C). In clear contrast, mice that were infected with PbA at later time points after BCG vaccination (after 70 or 130 days) did not show improved survival when compared to unvaccinated controls and the majority of animals in both groups developed ECM between 8 to 10 days.

We next investigated whether improved survival after BCG vaccination was associated with enhanced blood brain barrier (BBB) integrity. To do so, Evans blue was injected i.v. at day 8 post PbA infection. 1 h later mice were sacrificed and brains dissected. Brains of PbA-infected mice displayed equally distributed dark blue staining of the tissue, indicative of vascular leakage, whereas brains of mice that had received BCG 30 days before PbA inoculation only displayed faint staining (Figure 1D). As a control, brains of mice that only received BCG displayed no staining. In conclusion, our data suggest that BCG-mediated protection from ECM is associated with improved BBB integrity.

To get an impression whether BCG mediates its protective effects via affecting PbA liver- or blood-stage infection, we performed one experiment where mice were challenged with infected red blood cells (iRBC) instead of SPZ 30 days after BCG immunization. Again, the majority of BCG immunized mice did not develop clinical symptoms of ECM while all non-immunized mice developed ECM and succumbed to PbA infection by day 8 (Figure 1E). Together our data indicate that BCG mediates protection from ECM by interfering with PbA blood-stage infection.

### 3.2. Protection from ECM Is Associated with the Recovery of Viable BCG

Because BCG-mediated protection from ECM waned over time, we wondered whether BCG must be present in order to exhibit its protective effects. BCG was recovered from the vaccine draining lymph node up to 10 days after immunization but not later (Figure 2A). Moreover, we were able to recover viable BCG from spleen and liver 10 and 30 days after immunization, respectively (Figure 2A), but we could not recover any bacilli from both organs 70 and 130 days after immunization, indicating a link between the presence of viable BCG and protection from ECM. To substantiate these findings, we next vaccinated mice with heat-killed BCG 30 days before PbA challenge. As shown in Figure 2B, heat-killed BCG did not protect from ECM, corroborating our conclusion that viable BCG is needed to provide protection from ECM.

### 3.3. BCG Vaccination Does Not Lead to a Reduction in Parasite Load

Several studies have shown that the modulation of parasite burden protects against ECM [28,29,30,31,36]. This might at least in part be explained by the reduced availability of parasite antigen in the brain microvasculature [37]. Because BCG was reported to limit parasitemia in mice [11,12,17], we next asked if the protection from ECM could be explained by a lower blood-stage parasitemia in BCG-vaccinated mice. Importantly, blood-stage parasitemia is not only reduced by an improved blood-stage directed immunity but also as a result of improved elimination of pre-erythrocytic stages. Therefore, we compared pre-erythrocytic and blood-stage parasite development after PbA SPZ infection in the presence and absence of BCG. To our surprise, neither the parasite load in the liver 42 h after PbA infection nor blood-stage parasitemia differed between BCG-vaccinated or unvaccinated mice (Figure 2B,C). Even more surprising was our observation that BCG-vaccinated mice that did not succumb to ECM had a higher parasite load in the brain when compared to those mice that succumbed to ECM (Figure 2D). This indicates that BCG-mediated protection from ECM is not due to anti-parasite effector mechanisms and that parasite sequestration in the brain is not sufficient to induce ECM.

### 3.4. BCG-Mediated Protection from ECM Is Associated with Reduced Pro-Inflammatory Mediators in the Brain

Because parasitemia and parasite sequestration was not reduced in BCG-vaccinated mice, we hypothesized that an altered immune environment in these mice might explain the ablation of ECM. There is a growing consensus that overexpression of pro-inflammatory cytokines, including TNF, lymphotoxin α (LTα), and IFNγ, significantly contributes to the pathogenesis of the disease [38]. These cytokines likely promote the activation of the brain endothelium, the recruitment of leukocytes, the sequestration of infected and non-infected red blood cells and BBB damage [39,40]. We measured expression of these cytokines by qPCR (Figure 3A) and found the gene expression of Tnf, Ifnγ and Ltα to be significantly reduced in the brains of BCG-vaccinated PbA-infected mice in comparison to those infected in the absence of BCG (Figure 3A).

Expression of the anti-inflammatory cytokine Il10 was also significantly lower in BCG-vaccinated mice, which probably reflects the overall reduction in inflammation in brains of those animals. A hallmark of endothelial activation during ECM is the expression of adhesion molecules such as VCAM-1 and ICAM-1 on the endothelial cell surface. In good agreement with decreased pro-inflammatory responses, gene expression of adhesion molecules Vcam1 and Icam1 was significantly lower in brains of BCG-vaccinated compared to those of unvaccinated mice (Figure 3A).

Moreover, multiplex analysis revealed reduced production of relevant chemokines such as CXCL1, CXCL9, CXCL10, CCL2, CCL3 and CCL4 after PbA infection when mice had been vaccinated with BCG before (Figure 3B). In conclusion, our data suggest that activation of the brain endothelium and brain recruitment of immune cells involved in the pathogenesis of ECM is decreased when mice received BCG vaccination.

### 3.5. BCG Vaccination Reduced Cellular Influx into the Brains upon PbA-Infection

Reduced expression of chemokines and adhesion molecules in BCG-vaccinated mice indicated reduced recruitment of leukocytes to the brain. In ECM, accumulating T cells have major pathogenic roles. While CD4^+^ T cells are required during the induction phase of ECM and support the accumulation of pathogenic CD8^+^ T cells in the brain [41], parasite-specific CD8^+^ T cells cause local damage to the brain endothelium by the release of perforin and granzyme B (GZMB), causing BBB breakdown and neurological damage [31,42]. Reduced levels of CXCL9 and CXCL10 indicated impaired recruitment of CXCR3^+^ T cells to the brains of BCG-vaccinated mice. Indeed, flow cytometry revealed a significant reduction in the percentage of CXCR3^+^CD8^+^ T cells in the brain 8 days after PbA infection when mice had received BCG 30 days before (Figure 4A). These data were corroborated by lower expression of Cd8a in brains of ECM-free BCG-vaccinated animals compared to those that developed ECM after PbA infection in absence of BCG (Figure 4B). Moreover, gene expression of Gzmb in the brain tissue of BCG-vaccinated mice was very low compared to non-vaccinated PbA-infected mice (Figure 4B). Our data indicate that BCG dampened infection-induced inflammatory processes in the brain of PbA-infected mice, thereby interfering with the recruitment of detrimental effector cells to the brain, which prevents the onset of ECM.

### 3.6. BCG Changes T Cell Phenotype in Blood and Spleen Prior to the Onset of ECM

The differences in inflammation and infiltrating T cells in brains of BCG-immunized compared to non-immunized PbA-infected mice prompted us to analyze T cell phenotypes in blood and spleen on day 6 post PbA infection in order to reveal potential differences shortly before the onset of ECM symptoms. Like in the brain, the proportion of CD8^+^ T cells that expressed CXCR3 was significantly reduced in the blood, but not the spleen (data not shown) of mice that had received BCG prior to PbA infection (Figure 5A). This indicates that BCG prevents the recruitment of CXCR3^+^CD8^+^ T cells from the spleen, the site where immune effector cells to blood-stage infection are primed, via the blood into the brain. Further investigations also revealed a significant reduction in the expression of CD160 on CD8^+^ effector T cells (CD44^+^CD62L^−^) in the blood (Figure 5A). Expression of CD160 was shown to be specifically induced in highly activated, cytotoxic CD8^+^ T cells concurrently with the onset of cerebral symptoms [43].

One possible explanation why BCG vaccination leads to blunted T effector cell responses in response to PbA infection could be the induction of inhibitory receptors. We indeed found a significantly higher proportion of CD8^+^ effector T cells in blood and spleen of mice that had received BCG before PbA infection to express PD1, and moreover, these PD1^+^CD8^+^ T cells showed significantly higher median fluorescence intensity (MFI) for PD1, suggesting increased PD1 expression per cell (Figure 5A,B). In summary, our data suggest that the prevention of ECM in PbA-infected mice seems to be associated with anti-inflammatory and T cell inhibitory functions of BCG.

### 3.7. BCG-Induced Immunomodulation Is Lost When PbA Infection Was Performed after 130 Days

Since BCG-induced protection from ECM waned over time, we wondered whether loss of protection was related to the expression of pro-inflammatory cytokines and the presence of CD8^+^ T cells in the brain. In contrast to reduced expression of Ifnγ and Tnf in mice that were vaccinated with BCG 30 days before PbA challenge (Figure 3A), we could not detect a difference in the expression of both cytokines when mice were vaccinated with BCG 130 days before PbA challenge (Figure 6A). Likewise, we did not observe differences in CXCR3^+^CD8^+^ T cells recruited to the brain (Figure 6B). Next, we analyzed the expression of PD1 on CD8^+^ effector T cells in blood and spleen. Both the proportion of PD1^+^ T cells and the MFI of PD1 on CD8^+^ T cells was only slightly but not significantly elevated in BCG-vaccinated animals. Together these data corroborate our conclusion that a reduction in infection-induced inflammatory processes were responsible for the observed short-term protection mediated by BCG.

## 4. Discussion

Evidence has emerged to suggest that the antituberculosis vaccine BCG may offer beneficial off-target effects, providing some protection against diseases other than TB [44]. In the present study, we used a rodent model to show that a single s.c. immunization with BCG partially protects against ECM. However, opposite to our original hypothesis that BCG-induced trained immunity would protect mice from ECM by reducing parasitemia, BCG-mediated protection from ECM was not associated with a reduction in parasite burden, suggesting that BCG did not improve anti-parasite effector mechanisms. In line with this finding, pro-inflammatory mediators were not elevated but decreased in BCG-vaccinated animals, indicating immunosuppressive rather than immune activating functions of BCG.

Although BCG is a strong inducer of a Th1 type immunity, an immunosuppressive activity of BCG was already described 40 years ago [45,46]. These early studies suggested that BCG modified the lymphocyte compartment and activates natural suppressor cells in the bone marrow. Today, epidemiological studies suggest that BCG vaccination reduces the risk of diseases in which inflammation plays an important role, such as asthma, multiple sclerosis (MS) or Type 1 diabetes [47]. For instance, BCG vaccination reduced the frequency of active lesions in the central nervous system of MS patients and the risk of developing clinically definite MS for 5 years [48,49]. In line with these observations, BCG attenuated the severity of experimental autoimmune encephalomyelitis, a widely used mouse model of MS [50]. Moreover, BCG had neuroprotective effects in experimental models of neurodegenerative diseases [51,52,53] where it induced anti-inflammatory and inflammation-resolving effects by the recruitment of regulatory T cells or inflammation-resolving monocytes to the brain. Together these studies clearly indicate that BCG indeed may protect against inflammatory conditions and are in line with a recent study which shows that BCG vaccination in humans inhibits systemic inflammation [54]. Excessive inflammatory immune responses during infections with *Plasmodium* parasites are associated with severe complications such as CM. Specifically, pro-inflammatory cytokines such as LTα and IFNγ drive the development of ECM during PbA infection, as mice with either LTα or IFNγ deficiency are completely resistant to ECM [55,56,57]. IFNγ promotes the up-regulation of adhesion molecules such as ICAM-1 on brain endothelial cells and the expression of CXCL9 and CXCL10, thereby facilitating the migration and accumulation of pathogenic CD8^+^ T cells within the brain of PbA-infected mice [41,58]. In our study, reduced expression of IFNγ, LTα, ICAM-1, CXCL9 and CXCL10 were in line with impaired recruitment of CXCR3^+^CD8^+^ T cells to the brains of mice that received BCG before PbA infection. Moreover, we detected a significant expression of *Gzmb* in response to PbA infection in brains from previously naive but not from previously BCG-vaccinated mice. Thus, our data suggest that BCG immunization prevented inflammatory processes including the recruitment of pathogenic CD8^+^ T cells within the brain of PbA infected mice and thereby the development of ECM.

BCG-induced protection was associated with the induction of the inhibitory receptor PD1 on CD8^+^ T cells. PD1 often shows high and sustained expression levels during persistent antigen encounter [59]. Since the protective effect of BCG was linked to the presence of viable bacteria in the spleen, the main organ involved in the development of the immune response during blood-stage malaria [60], persistent BCG-derived antigen stimulation could be responsible for the up-regulation of PD1 on CD8^+^ T cells in spleen and blood. PD1 has an essential role in balancing protective immunity and immunopathology and is a marker of functional T cell exhaustion [59]. While this could be detrimental for control of chronic infections or cancer, it could be beneficial in inflammatory conditions where immunopathology needs to be restricted. Accordingly, C57BL/6 mice deficient in PD1 showed earlier neurological signs of ECM and shorter survival than WT mice while enhancing the PD1/PDL1 pathway using a PDL1 fusion protein improved the survival rate of PbA-infected mice via direct inhibition of CD8^+^ T cell functions [61]. We do not know the antigen-specificity of the PD1^+^CD8^+^ T cells in our study, but PD1 overexpression induced by BCG might still affect the availability and functionality of CD8^+^ T cells involved in the pathogenesis of ECM.

In our model, the protective effect of BCG waned over time and was associated with the recovery of viable BCG, indicating a link between the presence of BCG and protection from ECM. This observation is in line with an old report from Smrkovski in 1981 who described that the administration of BCG, irrespective of route, induced short- but not long-term protection against subsequent PbA challenge [14]. Another report from 1978 reported that the BCG-induced resistance against infection with *Schistosoma mansoni* was found to last for eight weeks and was dependent on the presence of significant numbers of viable organisms [62]. Likewise, infection with viable but not heat-killed BCG significantly increased the resistance to *Listeria monocytogenes* within the first two weeks but rapidly declined thereafter [63]. The reasons why the observed heterologous protection required viable BCG remain elusive. On the one hand, lack of heterologous protection could be a consequence of reduced amount of mycobacterial antigen for either innate or adaptive stimulation due to the inability of heat-killed bacilli to replicate or to disseminate to distant organs such as the spleen. This could also explain why protection wanes as soon as viable BCG are no longer detectable and the antigen load declines. On the other hand, secretory proteins which are absent after vaccination with killed bacteria might play an important role in unspecific protection.

Persistence of viable BCG was rather short in our model. Other studies have shown that BCG can persist in mice for much longer [64,65], and we do not know how long BCG persists in humans. The differences observed in murine studies might be due to differences in the BCG strain used, the route of application or the amount of bacilli administered. In addition, the genetic background of the mice can play a role. For instance, one study showed that BCG persisted at higher levels in spleens of BALB/c then C57BL/6 mice [66]. Therefore, the BCG-induced protection from ECM might last longer depending on the vaccine strain used and the genetic background of the recipient.

Epidemiological studies also imply that the heterologous BCG effects manifest soon after immunization, but wane over time and may be the most apparent for neonates vaccinated at birth [3]. Very recently, a large, multinational study suggested that BCG vaccination is associated with a reduced risk of malaria in children under the age of 5 years in sub-Saharan Africa [10]. The association was largest in regions with suboptimal BCG coverage. This study clearly implies that timely BCG vaccination could aid the global efforts to reduce malaria burden and emphasizes the need to improve BCG vaccination practice in regions with suboptimal coverage. Importantly, if BCG was able to prevent CM as our data suggest, this established vaccine might serve a feasible, safe and cheap approach to prevent fatal malaria cases particularly in areas with reduced malaria transmission. To circumvent limitations related to the requirement of persistent live bacilli, understanding the exact mechanisms behind the immunoprotective effects of BCG can provide the rationale for developing new effective adjunctive therapies to reduce the risk of death and brain damage in CM.

## 5. Conclusions

Our study indicates that BCG can prevent cerebral malaria by immunosuppressive mechanisms that are independent from the concept of trained immunity.

## Figures and Tables

**Figure 1 vaccines-08-00745-f001:**
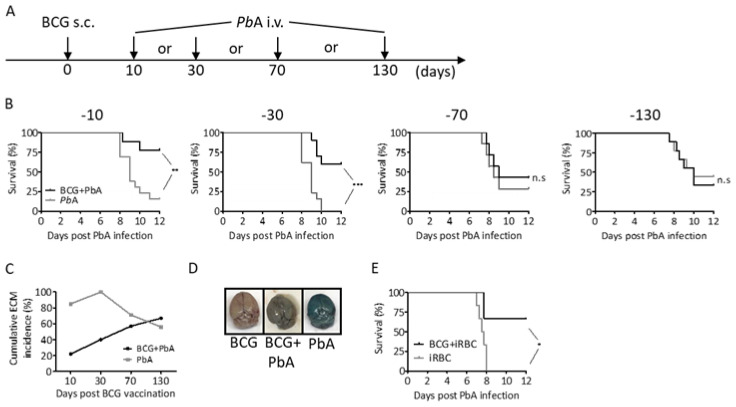
ECM development in previously naive or BCG-vaccinated mice. (**A**) Mice were s.c. immunized with 10^7^ BCG followed by i.v. challenge with 10^4^ PbA SPZ at different time points after BCG inoculation. (**B**) ECM-free survival in BCG immunized versus untreated (non-immunized) control animals following challenge with 10^4^ PbA SPZ i.v. (data from two independent experiments are shown, respectively; n = 9–13 for −10; n = 10–13 for −30; n = 7 for −70; n = 9 for −130; ** *p* < 0.01; *** *p*  < 0.001; n.s. = not significant; log rank test). (**C**) Cumulative ECM incidence (%). (**D**) Evans blue staining of brains on day 8 of PbA infection (representative images from one out of five mice per group are shown, respectively). (**E**) Survival of previously naive or BCG-immunized mice after challenge with PbA-iRBC (n = 6; * *p* < 0.05; log-rank test).

**Figure 2 vaccines-08-00745-f002:**
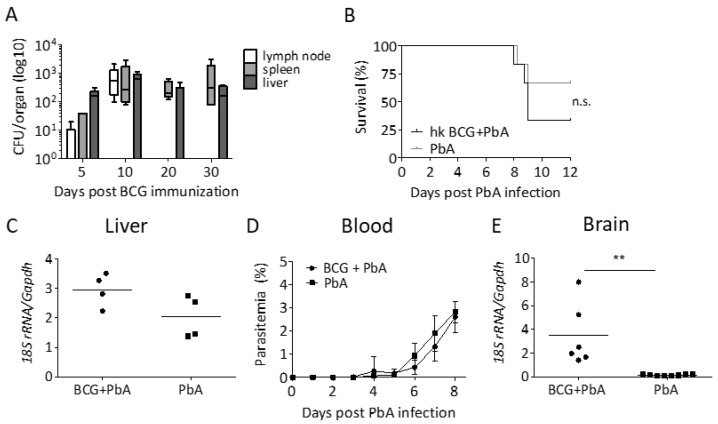
Bacterial and parasitic load. (**A**) Mice were immunized s.c. with 10^7^ BCG and at the indicated times, CFU was determined in lymph nodes, spleen, and liver (n = 5). (**B**) ECM-free survival in mice that were immunized with heat-killed (hk) BCG or left untreated and challenged with 10^4^ PbA SPZ i.v. 30 days later (n = 6; log-rank test; n.s. = not significant). (**C**–**E**) Naive mice or mice that had been vaccinated with BCG 30 days before were infected with PbA SPZ. Liver load was determined 42 h after PbA infection (**C**). Data from one out of two independent experiments are shown; symbols and bars represent individual mice and means, respectively. (**D**) Parasitemia was determined on Giemsa-stained blood smears from tail blood. Data are shown as mean ± SD from one representative experiment out of three; n = 6. (**E**) Parasite load in brains of mice that succumbed to ECM and those that were protected by BCG immunization (data from one out of two independent experiments are shown; symbols and bars represent individual mice and means, respectively). Statistical analysis was performed by Student’s *t*-test; ** *p* < 0.01.

**Figure 3 vaccines-08-00745-f003:**
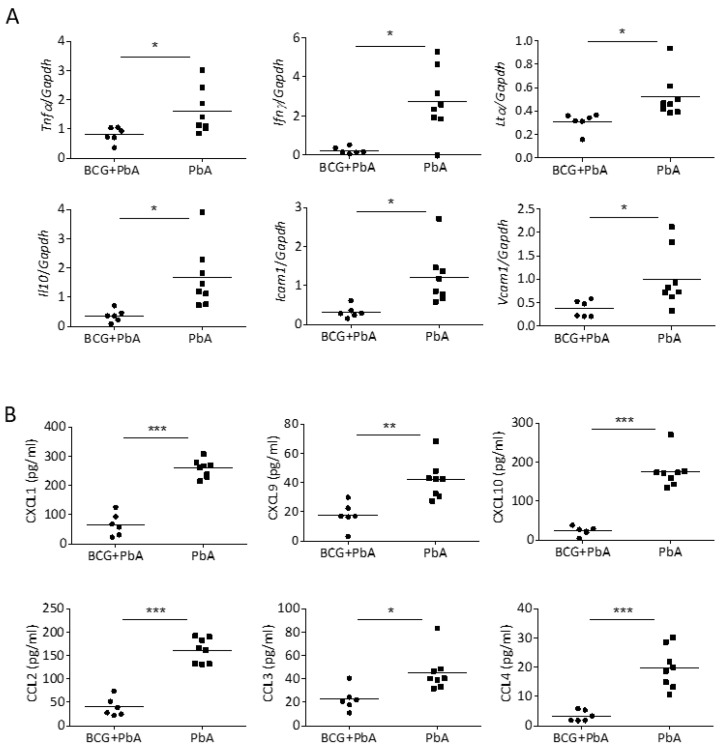
Assessment of immune mediators in the brain. Brains of mice that were or were not immunized with BCG 30 days before PbA SPZ challenge were analysed for the expression of cytokines or adhesion molecules (**A**) by qRT-PCR or for the protein levels of different chemokines by Legendplex (**B**). Data from one of two independent experiments are shown. Symbols and bars represent individual mice and means, respectively (n = 6–8). * *p* ≤ 0.05; ** *p* ≤ 0.01; *** *p* ≤ 0.001 determined by Student’s *t*-test.

**Figure 4 vaccines-08-00745-f004:**
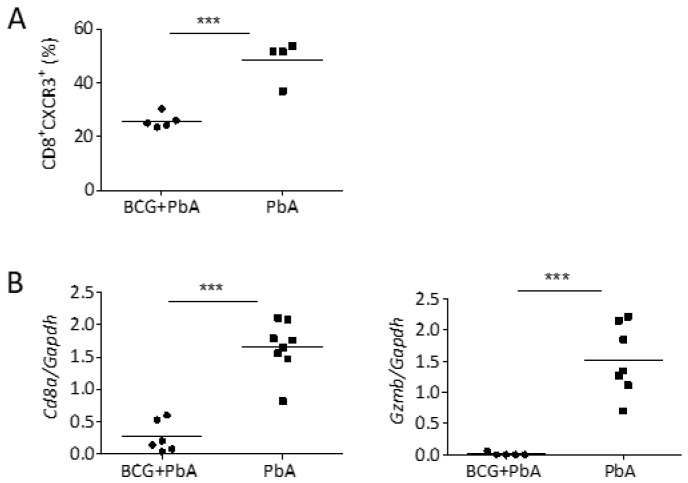
Assessment of CD8^+^ T cell responses in brain. (**A**) Flow cytometric assessment of CXCR3^+^CD8^+^ T cells in the brain on day 8 post PbA infection in previously naive or BCG-vaccinated mice (n = 4–5). (**B**) Brains of mice that were or were not immunized with BCG 30 days before PbA SPZ challenge were analyzed for the expression of Cd8a and Gzmb by qRT-PCR (n = 6–8). Data from one of two independent experiments are shown. Symbols and bars represent individual mice and means, respectively. *** *p* ≤ 0.001 determined by Student´s *t*-test.

**Figure 5 vaccines-08-00745-f005:**
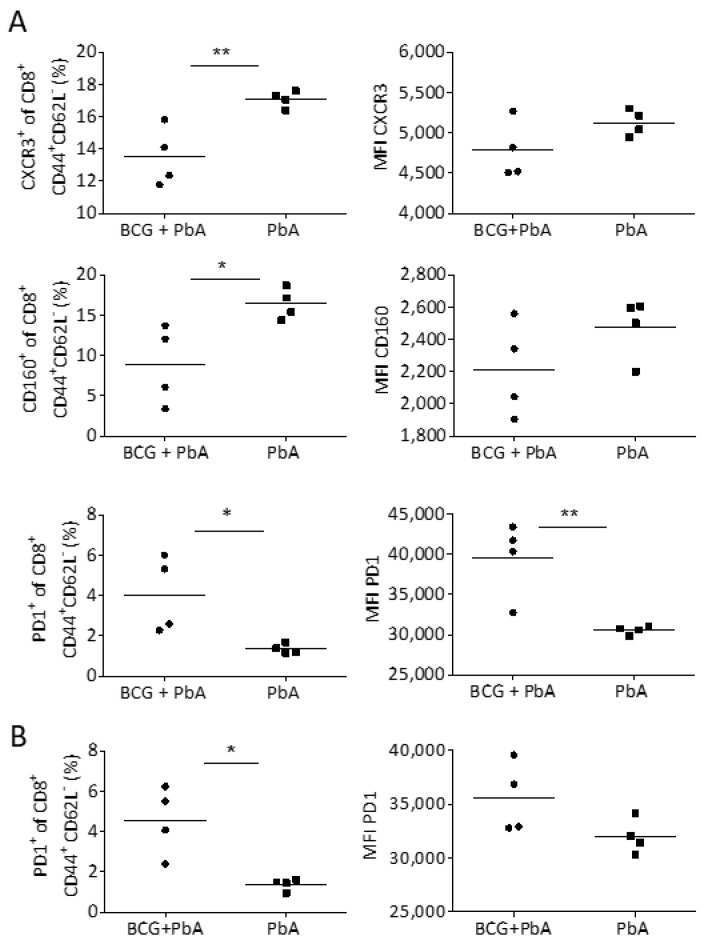
Assessment of CD8^+^ T cell phenotype in blood and spleen. Flow cytometric assessment of the activation status and surface expression of CXCR3, CD160 or PD1 on CD8^+^ T cells in blood (**A**) or spleen (**B**) on day 6 after PbA infection in previously naive or BCG-vaccinated mice (n = 4). Symbols and bars represent individual mice and means, respectively. * *p* ≤ 0.05; ** *p* ≤ 0.01 determined by Student’s *t*-test.

**Figure 6 vaccines-08-00745-f006:**
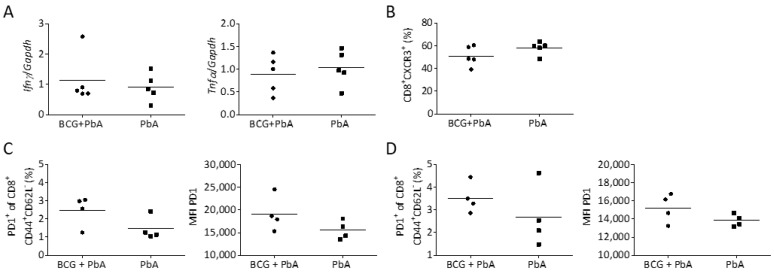
Assessment of immune responses in brain, blood and spleen in response to PbA infection 130 days after BCG vaccination. Mice were vaccinated with BCG or left untreated and challenged with PbA 130 days later. (**A**) Expression of cytokines by qPCR and (**B**) flow cytometric assessment of CXCR3^+^CD8^+^ T cells in the brain on day 8 post PbA infection. (**C**,**D**) Flow cytometric assessment of the surface expression of PD1 on CD8^+^ effector T cells in blood (**C**) or spleen (**D**) on day 6 after PbA infection. Symbols and bars represent individual mice and means, respectively. n = 5 (**A**,**B**) or 4 (**C**,**D**). Statistical analysis was performed by Student´s *t*-test.

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
