# Peer review of "BCG Provides Short-Term Protection from Experimental Cerebral Malaria in Mice"

_vaccines, 2020, doi:10.3390/vaccines8040745_

Round 1

Reviewer 1 Report

In the manuscript “BCG vaccination partially protects mice from experimental cerebral malaria” Witschowski et al. present their work on how cerebral malaria in mice can be prevented if mice where vaccinated by BCG in a short time frame before the encounter of the parasite. They carefully elucidate the underlying mechanisms that are independent on parasite burdens but instead associate with a reduced immune activation, less leaky BBB and reduced immune cell infiltration into the brain. These findings are of importance for the field as they indicate possible prevention mechanisms of cerebral malaria and are independent on the initial hypothesis of trained immunity by BCG. I have a few comments that I would like to see discussed or experimentally addressed to strengthen some statements of the authors:

  1. The BCG effects last around the same time as bacteria are present and the authors therefore conclude that live bacteria are needed. However, I am not convinced that this is a live BCG-specific effect but could also be provided by stimulation with heat-killed BCG? Does the effect really have to be antigen dependent? How about a role for innate stimulation by eg. ManLAM, Pam3 as possibilities?
  2. Nitric oxide that is also IFNg inducible is well known for its vasodilating role and to induce tissue damage. Are any differences in iNOS apparent in the brain pcr (fig 3)? Activated matrix metalloproteinases (MMP) are known to facilitate white blood cell invasion into the brain parenchyma. Are differences apparent in MMP expression?
  3. The last conclusion in the discussion “Importantly, if BCG was able to prevent CM as our data suggest, this established vaccine might serve a feasible, safe and cheap approach to prevent fatal malaria cases particularly in areas with reduced malaria transmission” is not completely clear to me and might be an overstatement as the authors do not show any long lasting effects. Do the authors have any suggestions to prolong the BCG-induced protection. Do I understand correctly that the authpors suggest BCG vaccine as a post malaria exposure treatment to prevent CM? Please clarify in discussion.
  4. The title could be improved to specify that effect on CM is short after BCG vaccination.

Author Response

In the manuscript “BCG vaccination partially protects mice from experimental cerebral malaria” Witschowski et al. present their work on how cerebral malaria in mice can be prevented if mice where vaccinated by BCG in a short time frame before the encounter of the parasite. They carefully elucidate the underlying mechanisms that are independent on parasite burdens but instead associate with a reduced immune activation, less leaky BBB and reduced immune cell infiltration into the brain. These findings are of importance for the field as they indicate possible prevention mechanisms of cerebral malaria and are independent on the initial hypothesis of trained immunity by BCG. I have a few comments that I would like to see discussed or experimentally addressed to strengthen some statements of the authors:

  1. The BCG effects last around the same time as bacteria are present and the authors therefore conclude that live bacteria are needed. However, I am not convinced that this is a live BCG-specific effect but could also be provided by stimulation with heat-killed BCG? Does the effect really have to be antigen dependent? How about a role for innate stimulation by eg. ManLAM, Pam3 as possibilities?

We thank the referee for bringing up this important point. Indeed, we also tested vaccination with heat-killed BCG, but we did not observe significant differences in the ECM rate between the treated and the untreated animals. Unfortunately, the overall ECM rate was rather low in this experiment, however, since even more mice that received heat-killed BCG developed ECM when compared to those that were previously naive, we believe that the data still suggest that BCG needs to be viable for the observed protective effects. We have added a graph to Figure 2 (2B), describe the data in lines 211-214 and also discuss our observations and thoughts in more detail in lines 413-421.

  1. Nitric oxide that is also IFNg inducible is well known for its vasodilating role and to induce tissue damage. Are any differences in iNOS apparent in the brain pcr (fig 3)?

We analysed the expression of NOS2 in the brains of vaccinated and unvaccinated mice, however our results were inconclusive. In one experiment, NOS2 expression did           not differ between the groups, while in a second experiment it was significantly lower in brains of BCG-vaccinated mice. Still, in both experiments, IFNg expression was significantly reduced in brains from BCG-vaccinated mice. We decided not to show data on NOS2 expression because they suggest that it is most likely not a major factor that contributes to the observed protective effects of BCG in our model. If the referee whishes to see the data or wants us to discuss them in the manuscript we are of course happy to do so.

  1. Activated matrix metalloproteinases (MMP) are known to facilitate white blood cell invasion into the brain parenchyma. Are differences apparent in MMP expression?

We tested the expression of MMP9 in brains but did not find a significant difference between BCG-vaccinated and non-vaccinated mice. Therefore, we did not show or discuss the data, but can of course do so if that is preferred by the referee or the editor.

  1. The last conclusion in the discussion “Importantly, if BCG was able to prevent CM as our data suggest, this established vaccine might serve a feasible, safe and cheap approach to prevent fatal malaria cases particularly in areas with reduced malaria transmission” is not completely clear to me and might be an overstatement as the authors do not show any long lasting effects. Do the authors have any suggestions to prolong the BCG-induced protection. Do I understand correctly that the authpors suggest BCG vaccine as a post malaria exposure treatment to prevent CM? Please clarify in discussion.

We thank the referee for bringing this to our attention. In fact, persistence of BCG was rather short in our model, and other studies have shown that BCG can persist in mice for much longer. Moreover, we do not know how long BCG does persist in vaccinated humans. Therefore, the BCG-induced protection might last longer depending on the vaccine strain used and the genetic background of the recipient. We do not suggest to use the BCG vaccine as a post malaria exposure treatment as we have not tested this experimentally yet. We have amended the discussion to make this more clear (please see lines 422-428; 437-438).

The title could be improved to specify that effect on CM is short after BCG vaccination.

We thank the referee for this suggestion and we changed the title accordingly.

Reviewer 2 Report

The present paper by Witschkowski et al  presents data suggesting that BCG vaccination before parasite challenge can protect mice from cerebral malaria. The  animal model, experimental procedure and results are sound and support their conclusions that BCG induces protection through induction of anti-inflammatory mechanisms, involving PD-1 and reduced expression of several proinflammatory genes.

Specific comments:

The protective BCG effect diminishes with time. How does this effect relate to the number of CD8+T cells in blood and brain and expression of PD-1 and proinflammatory cytokines?

Does BCG also affect T-reg T cells, which could be part of the protective effect.

It is well known that epigenetic changes  and trained immunity are induced  by BCG. It could be discussed why these changes do not affect malaria mortality in this model.

RBC rosetting is one pathogenic mechanisms in cerebral malaria. Is this reaction absent in BCG-treated animals? 

Author Response

The present paper by Witschkowski et al  presents data suggesting that BCG vaccination before parasite challenge can protect mice from cerebral malaria. The animal model, experimental procedure and results are sound and support their conclusions that BCG induces protection through induction of anti-inflammatory mechanisms, involving PD-1 and reduced expression of several proinflammatory genes.

Specific comments:

The protective BCG effect diminishes with time. How does this effect relate to the number of CD8+T cells in blood and brain and expression of PD-1 and proinflammatory cytokines?

We have added new data on CD8+ T cells and pro-inflammatory cytokines in brains of mice that were no longer protected by BCG vaccination when applied 130 days before PbA challenge. In line with loss of heterologous protection, we could not observe differences in the amount of CD8+ T cells nor in the expression of IFNg and TNFa in presence or absence of BCG. Likewise, we only observed slight but statistically not significant differences in the expression of PD-1. A new Figure 6 has been added and data are described in the text in lines 330-340 .

Does BCG also affect T-reg T cells, which could be part of the protective effect.

Indeed BCG is known to induce Tregs, which we have mentioned in the discussion before (line 377). Unfortunately, we did not address the role of Tregs in the current study, but we hope to be able to do so in future studies.

It is well known that epigenetic changes and trained immunity are induced by BCG. It could be discussed why these changes do not affect malaria mortality in this model.

We described the concept of trained immunity in the intrduction and indeed, our original hypothesis was, that BCG would protect mice from ECM through a reduction of parasitemia due to the induction of innate immune effector mechansism (please see lines 41-51; 61-64). In the discussion, we stated „However, opposite to our original hypothesis that BCG-induced innate immune activation would protect mice from ECM by reducing parasitemia, BCG-mediated protection from ECM was not associated with a reduction in parasite burden, suggesting that BCG did not improve anti-parasite effector mechanisms.“ In the new version oft he manuscript, we replaced „BCG-induced innate immune activation“ by „BCG-induced trained immunity“ to make this more clear. We then discuss the immunosuprressive effects of BCG that have already been shown before and that are associated with the protection against ECM in our model.

RBC rosetting is one pathogenic mechanisms in cerebral malaria. Is this reaction absent in BCG-treated animals? 

We thank the referee for bringing up this important question. We did not analyse RBC rosetting in the current study, but we hope to be able to investigate if BCG does affect this process in the future.